# Feature-Assisted Machine Learning for Predicting Band Gaps of Binary Semiconductors

**DOI:** 10.3390/nano14050445

**Published:** 2024-02-28

**Authors:** Sitong Huo, Shuqing Zhang, Qilin Wu, Xinping Zhang

**Affiliations:** Institute of Information Photonics Technology, School of Physics and Optoelectronic Engineering, Beijing University of Technology, Beijing 100124, China; hst6110@emails.bjut.edu.cn (S.H.); qlwu2001@emails.bjut.edu.cn (Q.W.)

**Keywords:** machine learning, SISSO, insightful prediction, band gap, binary semiconductors

## Abstract

The band gap is a key parameter in semiconductor materials that is essential for advancing optoelectronic device development. Accurately predicting band gaps of materials at low cost is a significant challenge in materials science. Although many machine learning (ML) models for band gap prediction already exist, they often suffer from low interpretability and lack theoretical support from a physical perspective. In this study, we address these challenges by using a combination of traditional ML algorithms and the ‘white-box’ sure independence screening and sparsifying operator (SISSO) approach. Specifically, we enhance the interpretability and accuracy of band gap predictions for binary semiconductors by integrating the importance rankings of support vector regression (SVR), random forests (RF), and gradient boosting decision trees (GBDT) with SISSO models. Our model uses only the intrinsic features of the constituent elements and their band gaps calculated using the Perdew–Burke–Ernzerhof method, significantly reducing computational demands. We have applied our model to predict the band gaps of 1208 theoretically stable binary compounds. Importantly, the model highlights the critical role of electronegativity in determining material band gaps. This insight not only enriches our understanding of the physical principles underlying band gap prediction but also underscores the potential of our approach in guiding the synthesis of new and valuable semiconductor materials.

## 1. Introduction

Binary semiconductor materials play a vital role in many industries due to their unique physical properties. The importance of these materials stems largely from their band gap widths, a pivotal parameter that decisively influences the performance of various devices, such as field-effect transistors, photodiodes, photo transistors, and solar cells [1,2,3]. For instance, binary transition metal nitrides exhibit a wide range of physical properties, including superconductivity, metal-insulator transitions, ferroelectricity, and thermoelectricity characteristics, due to their large band gap distribution [4,5,6,7]. These properties have led to their widespread application in energy-related fields, such as energy storage, electrocatalysis, and photocatalysis. Particular attention has been given to third-generation semiconductor devices, especially GaN and AlN. These materials are known for their high breakdown electric field strength and wide band gaps [8], which gives them significant advantages in optoelectronic device fabrication, particularly in blue and ultraviolet light domains [9,10]. On the other hand, narrow band gap binary semiconductors like CdTe and InAs [2,11], with their high optical absorption coefficients and extended charge carrier lifetimes, are providing new insights for near-infrared active photoelectrodes [1]. This diversity in band gap ranges highlights the versatility of binary semiconductors, enabling them to serve different applications based on their specific band gap characteristics.

The evolution of high-throughput computing has unlocked a vast potential for exploring binary semiconductors [12,13]. However, the well-known ‘band gap problem’—the discrepancy between density functional theory (DFT) calculations and actual band gaps in semiconductors—remains a complex challenge in predicting band gap [14]. Advanced methods beyond standard DFT, such as HSE hybrid functionals, GW approximation, and DFT-embedded dynamical mean field theory, have been developed to provide more accurate electronic band structures. However, the high computational demands of these methods limit their practicality in high-throughput calculations [15].

Machine learning (ML) techniques, known for their precision and cost-effectiveness, have emerged as a novel approach, enabling materials scientists to predict the electronic properties of new materials by extracting and integrating information from extensive material databases [16,17,18]. For example, Xu et al. used ensemble learning models for the prediction of band gaps in thermoelectric materials with diamond-like structures, and they achieved a prediction accuracy of 77.73% [19]. Huang et al. used ML models trained on DFT results to accurately predict band gaps and alignment of nitride-based semiconductors [20].

A point of contention in the field has been the ‘black-box’ nature of ML models, which are often criticized for their limited ability to derive new physical laws, thereby constraining their potential in certain applications [21]. Symbolic regression, an approach that produces interpretable equations, is gaining attention in various scientific fields. A notable application of this approach is the sure independence screening and sparsifying operator (SISSO) method [22], which has been used successfully in band gap prediction models. For instance, Zhang et al. trained on high-throughput calculations of two-dimensional semiconductors and utilized complex descriptors identified by the SISSO algorithm, and they achieved high accuracy in predicting HSE band gaps with a coefficient of determination (*R*^2^) of 0.96 [23]. Ma et al. proposed a physically interpretable three-dimensional descriptor to obtain the Γ-point gap of twist bilayer graphene at arbitrary twist angles and different interlayer spacings, demonstrating high accuracy as evidenced by a 99% Pearson coefficient [24].

In this study, we developed an interpretable machine learning model for precise band gap prediction in binary semiconductors, using a limited but targeted set of elemental properties. This model provides in-depth insights into how specific descriptors affect the band gap, enhancing our understanding of the material’s intrinsic properties. We initially employed various conventional machine learning algorithms to train and optimize the model with non-metallic material data. The most effective models were then selected for interpretative learning. By assigning weights to the feature importance in these models, we identified the most effective feature combinations. Subsequently, the SISSO algorithm was applied to discern descriptors from the feature space that accurately represent the band gap. Utilizing the SISSO algorithm, we could generate prediction models in the form of equations, thus providing significant physical insights regarding descriptors on the band gap. Overall, by integrating the machine learning algorithms and the SISSO algorithm, we have established a model with strong interpretability and physical significance, which is a valuable tool and methodology for materials design and development. The transparency and comprehensibility of this approach hold broad applicative potential in the field of materials science.

## 2. Methods

### 2.1. Dataset and Features

In our study, the experimental band gaps were taken from the work of Ya Zhuo et al. [25]. We selected 1107 binary semiconductors (356 materials with different compositions) from their collection of experimental band gaps of 6354 inorganically stable materials. The DFT calculated band gaps based on PBE functional [26] were from the Materials Project (MP) database. For those with notable differences between DFT calculations and experimental band gaps, we have included multiple sets of experimental band gap values. This enhanced the model’s capacity to capture the compositional effects of these compounds, thereby constructing a machine learning model that better aligns with practical research requirements.

For the ML models, we selected 11 intrinsic features of the compound’s constituent elements that have been considered highly relevant to the band gap, as identified in the previous literature [19,27]. These features include electronegativity, first ionization energy, atomic mass, atomic number, column number, row number, period number, group number in the periodic table, ionic radius, atomic radius, and density at 25 °C. Additionally, the PBE-calculated band gap in the MP platform [13] was also used as an input feature. Therefore, each binary semiconductor material A_m_B_n_ has 23 input features: 11 intrinsic features for each of the two elements, as well as the calculated band gap provided by the MP database. Details of the features are shown in Table 1. The group number (period number) of non-rare earth elements is equal to their column number (row number). For rare earth elements, their group number is determined by adding 15 to their column number, and their period number is determined by subtracting 2 to their row number. This ensures that each element can be uniquely represented by a combination of period number and group number.

### 2.2. Evaluation Metrics

We employed the root-mean-square error (*RMSE*) and the *R*^2^ as primary metrics to evaluate the regression model. *RMSE* provides a measure of the average magnitude of errors between the predicted and actual values, while *R*^2^ indicates the proportion of variance in the dependent variable that is predictable from the independent variables, offering insight into the fit quality of the model. They are described as follows:$$R^{2}=1-\frac{\Sigma_{i=1}^{n}{\left({\hat{y}}_{i}-y_{i}\right)}^{2}}{\Sigma_{i=1}^{n}{\left(y_{i}-{\bar{y}}_{i}\right)}^{2}}$$
$$RMSE=\sqrt{\frac{1}{n}\sum_{i=1}^{n}{\left({\hat{y}}_{i}-y_{i}\right)}^{2}}$$
where *y_i_* is the experimental band gap, ${\bar{y}}_{i}$ is the corresponding average, ${\hat{y}}_{i}$ is the predicted band gap, and n is the number of samples. In order to avoid random errors, we used the calculated results of three-fold cross-validation methods to judge the predictive ability of the model.

We used the permutation importance method [28] from the scikit-learn library [29] to assess the importance of features in our models. This method is independent of the model itself. It works by randomly shuffling the values of each feature and observing how much this affects the model’s performance. A significant decrease in performance when a feature is shuffled indicates that the feature is important to the performance of the model. This approach helps us understand the impact of each feature on the model and ensures that the selected features contribute effectively to the model’s ability to generalize to new data.

## 3. Results and Discussion

### 3.1. Screening of Predictive ML Methods

For predicting band gaps in binary semiconductors, we evaluated six supervised ML techniques. Among them were two linear methods: LASSO, a linear regression method, and kernel ridge regression (KRR), which employs the kernel trick for non-linear data. We also examined the support vector regression (SVR) method, notable for its versatility with various kernel functions suitable for different data types. In addition, the decision trees (DT) method was appreciated for its simplicity and ease of interpretation. Furthermore, we considered two ensemble methods, random forests (RF) and gradient boosting decision trees (GBDT), both known for their robustness in complex predictive tasks. The modeling was implemented under the Python computing environment using the scikit-learn library [29].

Table 2 presents the performance of band gap prediction using six ML methods measured by two different evaluation metrics: *R*^2^ and *RMSE*. It was observed that the SVR model, RF model, and GBDT model showed lower *RMSE* values (*RMSE* < 0.4 eV) and larger *R*^2^ values (*R*^2^ > 0.950) for the train and test set, indicating that their predictions were closer to the target values. Therefore, the following analysis mainly focuses on the results of these three ML methods.

Figure 1 illustrates the predictive performance of ML methods in a single-shot random trial, utilizing a 90% training set (represented by blue dots) and a 10% test set (represented by yellow dots). Figure 1a shows a comparison between the experimental band gaps of materials and the PBE-calculated band gaps from MP, indicating a large deviation between PBE calculations and actual band gaps. Comparing the predicted band gaps of the SVR, RF, and GBDT models with the experimental values reveals a notable improvement in prediction accuracy of the ML models over the PBE calculations. Comparing the results of the three ML models, the RF model exhibits better fitting performance for the training set than the SVR model, but its accuracy in the test set is slightly lower. The GBDT model has a smaller overall deviation and provides more accurate predictions for wide band gap materials, but it shows a higher dispersion in predicting materials with smaller band gaps compared to the RF model.

To counter potential training inefficiencies from increased feature space complexity, we first refined the feature space by analyzing the feature importance of the SVR, RF, and GBDT results. This process included evaluating the impact of 23 input features for each model and then ranking them based on their importance. We averaged five measurements for a reliable feature importance ranking to reduce the impact of variability from multiple training. We observed common characteristics across the models, such as a high correlation between calculated and experimental band gaps, and significant importance weights for elements’ electronegativity. Figure 2 shows the top 14 most significant features of SVR, RF, and GBDT models used to form sub-training sets for the next step in searching for interpretable physical models.

### 3.2. Physical Insights from SISSO Predictions

The machine learning models mentioned above focus primarily on achieving the highest data prediction accuracy, which often leads to a reduction in the interpretability of the model. On the other hand, SISSO models strive to strike a balance between accuracy and complexity, enhancing both the understanding of the problem and the precision of predictions. As a compressed-sensing technique, SISSO is particularly adept at identifying the most efficient low-dimensional descriptor from a large pool of possible options. Utilizing the rankings of feature importance in Figure 2, we employed the SISSO method to derive an optimized descriptor for predicting the band gaps of binary semiconductors. Here, the input features that include the top 14 most significant features are considered for different models. Figure 3 presents the *RMSE* metric curves for one-dimension (1D) and two-dimension (2D) SISSO models, depicting how the *RMSE* varies with the number of input features. Here, the dimensionality refers to the count of fitting coefficients in a linear model, not including the intercept. For a 1D SISSO model, there is only one descriptor, denoted as *D*_1_, while a 2D model includes two descriptors, labeled *D*_21_ and *D*_22_. As the complexity of the features increases, the predictive ability of a SISSO model has a noticeable improvement in the initial stage. When the number of features increases to a certain extent, the *RMSE* of 1D models remains almost unchanged and those of 2D models only slightly decreases. The *RMSE* of SISSO model based on the GBDT ranking (SISSO-GBDT) becomes stable after the number of features reaches four. The *RMSE* of SISSO model based on the RF ranking (SISSO-RF) stabilizes after more than seven features, and the SISSO model based on the SVR ranking (SISSO-SVR) reaches a plateau in *RMSE* after exceeding 11 features. Using the top 4 features from GBDT, the top 7 from RF, and the top 11 from SVR as inputs for the SISSO method, we developed corresponding 1D and 2D physical models, as shown in Table 3. The correlation between the band gaps predicted by these models and the experimental band gap values is depicted in Appendix A. To further explore the predictive capabilities of the SISSO model, we focused on thermodynamically stable binary semiconductors with energy above the convex hull less than 10^−6^ eV/atom in the MP database [13], resulting in a dataset of 1208 distinct materials. This selection ensures that our study targets materials with demonstrated stability and relevance for practical applications. The prediction results of the three SISSO models on these 1208 materials are presented in Appendix A.

Considering both interpretability and accuracy, we have opted for the 1D representation of the SISSO model based on RF ranking (SISSO-RF) for a focused interpretation of its physical significance. Although this physical model incorporates a greater number of output features compared to the SISSO model based on GBDT ranking, its simplified version offers enhanced interpretability, making it our choice for further analysis. In this model, when the material system does not contain rare earth elements, the parameter C_A_/G_A_ equals 1. Consequently, *D*_1_ and the corresponding *E_g_* (SISSO-RF, 1D) should be the following:(1)$$D_{1}=E_{g.PBE}+\left|\left|E_{g.PBE}-A_{EN}\right|-\left|E_{g.PBE}-B_{EN}\right|\right|\;E_{g}=1.047\;\times D_{1}+0.134$$

In our study, we focused on binary semiconductors, denoted as A_m_B_n_, where ‘A’ represents a cation and ‘B’ represents an anion. It is generally observed that the element in a compound forming the anion is more electronegative than that forming the cation. This is because the anion needs higher electronegativity to attract electrons effectively. When *E_g.PBE_* < *A_EN_* or *E_g.PBE_* > *B_EN_*, the *D*_1_ term can be simplified to *E_g.PBE_* + (*B_EN_ − A_EN_*), which is defined as Δ. Figure 4a illustrates how the Δ varies with the *D*_1_ across different materials. It can be seen that in this case, where *D*_1_ is less than 2 eV or greater than 5 eV, that *D*_1_ and Δ are equivalent, so this simplification is reasonable and applicable for most materials we examined. When *D*_1_ is greater than 2 eV and less than 5 eV, i.e., in situations where the *E_g.PBE_* falls between A_EN_ and B_EN_ (*A_EN_* < *E_g.PBE_* < *B_EN_*), *D*_1_ should be simplified to |2*E_g.PBE_* − *B_EN_* − *A_EN_*|. The rationale behind these simplifications is that they reflect a specific relationship or characteristic within the semiconductor based on the electronegativity values. Moreover, for a material with known elemental types and band gap calculated by PBE, this approach allows for straightforward assessments and predictions. The effectiveness of our model is further demonstrated in Figure 4b, which shows a comparison between the band gaps predicted using our simplified formula and the experimentally determined band gaps. This comparison validates our model, confirming its applicability and accuracy in predicting band gap values in binary semiconductors.

Our band gap prediction model using the SISSO method, in contrast to the previous literature that relied solely on fitting formulas based on PBE gaps [30,31], additionally incorporates the parameter of electronegativity, making it more aligned with physical intuition [32]. First, the band gap represents the energy required for an electron to transition from the top of the valence band to the bottom of the conduction band, determined by the electron-attracting capacity of the bonding atoms, making the inclusion of electronegativity a logical choice for characterizing the material’s band gap. Using the descriptors provided by the SISSO model, we can adjust the elemental composition of materials to find semiconductors that meet specific band gap requirements. These SISSO descriptors are not only useful for statistically predicting the range of band gap energy values but also for assisting in understanding the mechanisms affecting the material’s band gap from various perspectives through machine learning. Additionally, the SISSO algorithm establishes quantifiable mathematical expressions that link these elemental properties with band gap descriptors, greatly aiding in our search for optoelectronic materials with suitable band gaps.

## 4. Conclusions

In this study, we developed a physically interpretable expression for accurately predicting the band gap of binary compounds using machine learning methods. To identify the optimal descriptors, our study adopted a segmented active learning approach. Initially, we used six machine learning regression models—including LASSO, KRR, SVR, DT, RF, and GBDT—to predict band gap values using PBE-calculated band gaps and 11 intrinsic elemental features. Among the six tested supervised machine learning models, SVR, RF, and GBDT showed superior performance. In the subsequent phase, we employed the SISSO approach, which utilized the feature importance rankings derived from the SVR, RF, and GBDT models. This led to the identification of specific descriptors by SISSO, which not only enabled precise predictions of the band gaps but also illuminated the fundamental factors influencing the band gaps in binary compounds. We then applied these descriptors to 1208 binary semiconductors in the MP database to predict the band gaps of these materials. In the final step, we further refined the SISSO model based on RF, discovering that for binary semiconductors without rare earth metals, the band gap prediction model only required a few key parameters, such as PBE band gaps and elemental electronegativity. These descriptors not only enhance our understanding of semiconductor materials but also open up new avenues for researching and discovering more ideal semiconductor materials.

## Figures and Tables

**Figure 1 nanomaterials-14-00445-f001:**
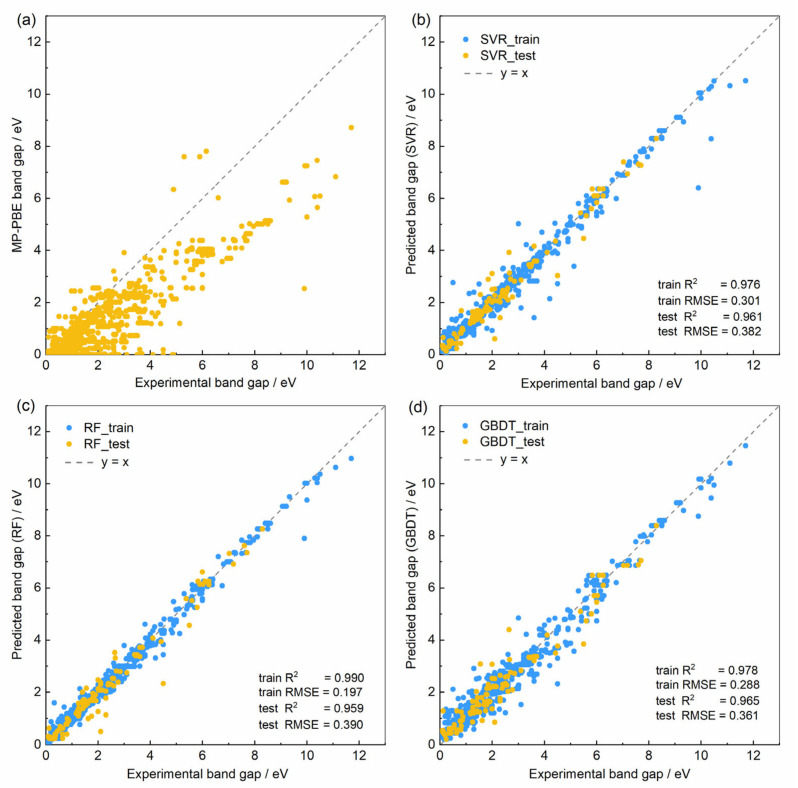
(**a**) Comparison of experimental band gap values with PBE-calculated band gaps from the MP materials database. (**b**–**d**) Comparison of experimental band gap values with the predicted band gap values by SVR, RF, and GBDT ML models, respectively.

**Figure 2 nanomaterials-14-00445-f002:**
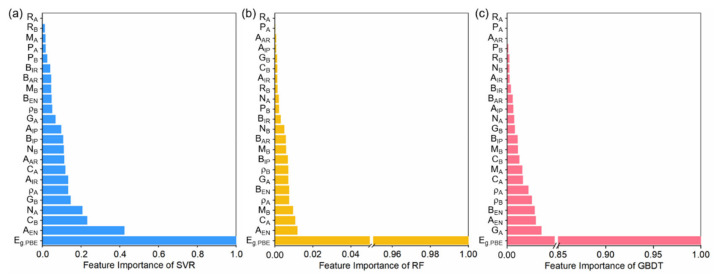
Feature-importance rankings and weights of the top 14 most significant features for the (**a**) SVR, (**b**) RF, and (**c**) GBDT models of band gap predictions. The feature importance weights of the models have all been normalized.

**Figure 3 nanomaterials-14-00445-f003:**
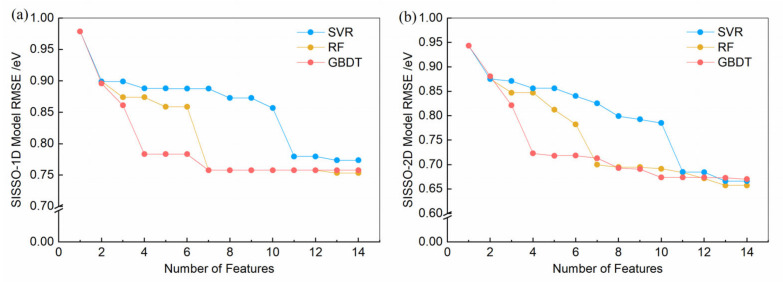
The *RMSE* of three SISSO models as a function of the number of input features, which are determined by the importance weight ranking results from the SVR, RF, and GBDT models: (**a**) 1D models and (**b**) 2D models.

**Figure 4 nanomaterials-14-00445-f004:**
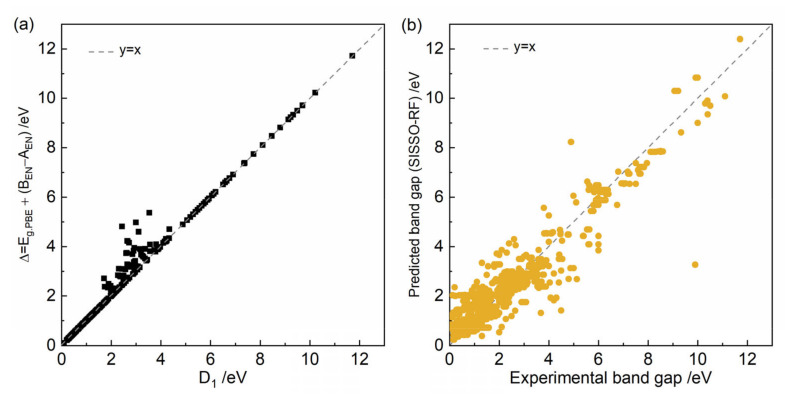
(**a**) The relationship between simplified Δ with *D*_1_. (**b**) Comparison between the band gaps predicted using our simplified formula and the experimentally determined band gaps.

**Table 1 nanomaterials-14-00445-t001:** Input features (descriptors) used in ML models for the purpose of predicting band gaps.

Symbol	Meaning in Binary Compound A_m_B_n_	Unit
A_EN_, B_EN_	electronegativity of A or B	eV
A_IP_, B_IP_	first ionization potential of A or B	eV
M_A_, M_B_	atomic mass of A or B	g/mol
ρ_A_, ρ_B_	density at 25 °C of A or B	g/cm^3^
A_AR_, B_AR_	atomic radius of A or B	Å
A_IR_, B_IR_	ionic radius of A or B	Å
N_A_, N_B_	atomic number of A or B	−
R_A_, R_B_	row number of A or B	−
C_A_, C_B_	column number of A or B	−
P_A_, P_B_	period number of A or B	−
G_A_, G_B_	group number of A or B	−

**Table 2 nanomaterials-14-00445-t002:** Evaluation metrics of six ML model on training and test sets.

ML Model	Training Set		Test Set	
	*RMSE* (eV)	*R* ^2^	*RMSE* (eV)	*R* ^2^
LASSO	0.717	0.864	0.727	0.857
KRR	0.442	0.948	0.535	0.922
SVR	0.301	0.976	0.382	0.961
DT	0.475	0.940	0.732	0.854
RF	0.197	0.990	0.390	0.958
GBDT	0.288	0.978	0.361	0.965

**Table 3 nanomaterials-14-00445-t003:** The 1D and 2D physical models using the SISSO method with the top 4 features from GBDT, the top 7 from RF, and the top 11 from SVR as inputs for each respective model.

Model	SISSO-SVR	SISSO-RF	SISSO-GBDT
1D	$E_{g}=0.057\times D_{1}-1.423$	$E_{g}=1.047\;\times D_{1}+0.134$	$E_{g}=\left(-0.862\right)\times D_{1}+0.332$
	$D_{1}=(2E_{g.PBE}+B_{IP})\times \frac{G_{B}-N_{A}}{A_{AR}\times A_{EN}}$	$D_{1}=E_{g.PBE}\times \frac{C_{A}}{G_{A}}+\left|\left|E_{g.PBE}-A_{EN}\right|-\left|E_{g.MP}-B_{EN}\right|\right|$	$D_{1}=A_{EN}-B_{EN}-E_{g.PBE}\;\;\;\;\;\;\;\;\;+\frac{A_{EN}-E_{g.PBE}}{exp\left(G_{A}\right)}$
2D	$E_{g}=-0.004\times D_{21}+4.340\times D_{22}-1.861$	$E_{g}=1.020\times D_{21}-0.047\;\times D_{22}-0.429$	$E_{g}=1.079\;\times D_{21}-3.276\;\times D_{22}+0.453$
	$D_{21}=\frac{{G_{B}}^{2}(A_{EN}-B_{FIP}-E_{g.MP})}{A_{R}\times A_{EN}}$	$D_{21}=\frac{\left(E_{g.MP}\times C_{A}\right)+\left(G_{A}\times B_{EN}\right)}{G_{A}\times {A_{EN}}^{3}}$	$\;D_{21}=\frac{E_{g.PBE}+B_{EN}}{{G_{A}}^{9}}$
	$D_{22}=\frac{E_{g.MP}}{N_{B}}+\frac{A_{EN}}{N_{A}\left|N_{B}-\left|N_{B}-G_{B}\right|\right|}$	$D_{22}=\frac{A_{EN}^{2}}{D_{A}\left|G_{A}/A_{EN}-M_{A}/B_{EN}\right|}$	$D_{22}=\left|\sqrt{\left(A_{EN}/B_{EN}\right)}-\frac{log(G_{A})}{exp(A_{EN})}\right|$

## Data Availability

The data presented in this study are available upon reasonable request from the corresponding author.

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
