# Peer review of "Feature-Assisted Machine Learning for Predicting Band Gaps of Binary Semiconductors"

_nanomaterials, 2024, doi:10.3390/nano14050445_

Round 1
Reviewer 1 Report
Comments and Suggestions for Authors
The subject of this study holds significant interest within its respective field. Although the methodology employed appears to lean towards statistical AI search, the outcomes and accuracy presented in this research are both reasonable and beneficial. I would be keen to see a more profound integration of advanced AI models with physical models in future work from this group, reflecting a higher level of expectation.
Author Response
Thank you very much for your positive feedback on our manuscript. Your suggestion regarding the integration of advanced AI models with physical models in future work aligns well with our vision for extending the scope and depth of our research. We are committed to addressing this aspect in our ongoing and future projects to meet the higher expectations you have expressed. Thank you again for your valuable comments and insights.
Reviewer 2 Report
Comments and Suggestions for Authors
The work is devoted to the development of machine learning algorithms for determining the band gap of binary semiconductor compounds.
Unfortunately, the work made a chaotic impression. The authors were unable to convey why this activity is being carried out and what practical results can be obtained by applying the discussed algorithms. The authors rely in their research on a database of experimental data and DFT calculations. At the same time, how new information about the band gap of a previously unknown material can come from remains unclear.
Below I have outlined my doubts about this work in the form of specific comments on the text.
Alas, I believe that the work cannot be published. The authors need to radically reconsider their approach to this work.
1. I will allow myself to express doubts about the relevance of the task. The authors use machine learning to fine-tune algorithms for calculating band gaps in binary semiconductor compounds. Although the periodic table gives us many options for the formation of binary semiconductor compounds, most of them have been well studied experimentally (especially the widely applicable III-V compounds, such as GaAs, InAs, GaN, AlN and many others). The their band gaps (and many other parameters, such as the lattice constant, Varshni parameters, etc.) are known with high accuracy. Moreover, the nonlinearity parameters of various solid solutions based on these compounds were studied experimentally. In this regard, it remains unclear for what other unstudied compounds the authors hope to apply the trained algorithms? The authors deliberately limited themselves to the class of binary semiconductor compounds, which gave rise to my question. If the authors show the applicability of their results to a wider class of materials, then this could explain the meaning of their work. In my opinion, such work can be valuable from the point of view of predicting the properties of new, previously unstudied compounds, which will inspire their synthesis and experimental study. On the other hand, perhaps the value of the authors’ work lies in optimizing the operation of specific algorithms for machine learning using a simple example of binary compounds. Algorithms tested on simple problems can be of practical use when applied to new, unknown materials. In this case, authors should shift the emphasis in the introduction to emphasize the significance of the work.
2. As I understand, the authors do not take into account the possibility of allotropic modifications of the same compounds. Let me explain what I mean: there is GaN with a sphalerite structure, and there is one with a wurtzite structure. These materials, despite the same chemical composition, have very different physical properties, including band gaps (3.299 eV for GaN in the zinc blende structure, and 3.5 eV in the wurtzite structure). There is an even more striking example - carbon (diamond, graphite, graphene), but it goes beyond binary compounds.
3. The meaning of yi with a line (the corresponding average, line 119) is absolutely unclear to me. How does averaging happen? Based on different experimental data or different PBE calculation results? In both cases, it is not clear where the scatter may come from (except for experimental error, which is usually taken into account as a +- gate when reporting such data). In the case of calculations, such an error is determined by the convergence criteria of a specific calculation algorithm.
4. It is not clear where from (other than PBE calculations) all the considered models obtain data on the bandgap value of each material for comparison with the experimental value. A detailed explanation is required.
5. line 238-239. Apparently these are technical edits that, for reasons unknown to me, remained in the final version of the text. This indicates clearly careless work with the text.
6. It is not clear what the meaning of parameters D1, D21 and D22 are. Some clarification required.
7. The authors mention different numbers of compounds used (1107 - line 88 and 1208 - line 273). Apparently this is a typo? Which value is correct?
8. It remains unclear exactly how the results obtained will help predict the band gap of the new material.
Reviewer 3 Report
Comments and Suggestions for Authors
In this manuscript, the authors perform machine learning (ML) to predict the bandgap of binary semiconductors. The emphasis is on combining different traditional ML algorithms with the 'white-box' sure independence screening and sparsifying operator approach. This is a useful idea entirely compatible with available semiconductor databases such as Materials Project database. Juxtaposing different (in this case three) Sure Independence Screening and Sparsifying Operator (SISSO) models is a productive idea too.
Novel semiconductor binaries have been studied before by ML schemes for both their structure and electronic properties but freshly developed ML models as the present one could improve bandgap prediction and subsequent experimental success.
The layout of the manuscript and the discussion of the results is well presented and easy to understand. Enough data is provided so most results can be seen as repeatable by other research teams with ML competence.
Some minor imperfections still preclude this manuscript from being acceptable for publication. It can be considered for publication after a minor revision:
1: Title: It is incorrect in English to begin each word in a title with a capital letter. Also, self-proclaiming wording like “insightful”, “novel” etc., are banalities, especially in a title. Therefore, such wording is better skipped.
2: Abstract: It would make the ML approach much clearer if any of the mentioned in the text features are explicitly mentioned in the abstract too. The reliance to different SISSO models should be mentioned.
3: Throughout the text, the relationship between thermal stabilities of the binary compounds and their eligibility to be included in the (ML) study is not discussed.
4. I recommend that the authors refer to the discussion of stability criteria, as in Matter 6, 2711 (2023). It is significant to include a detailed discussion on the stability criteria of the any considered material, being it in 3D or 2D, to justify its practical utility effectively.
5: The authors should illustrate the sampling of material systems they focus their ML modelling on by exemplifying newly discovered binary semiconductor structures such as InO (CrystEngComm 23 (2021) 6661-6667) that have been recently studied reliably and affordably by DFT.
6: A significant stylistic and grammatical revision of the manuscript is necessary.
Comments on the Quality of English Language
Spell-check and stylistic revision of the paper are necessary. Some long sentences, as well as misspellings, etc., are noticeable throughout the text.
Round 2
Reviewer 2 Report
Comments and Suggestions for Authors
The authors clarified the main issues and substantiated the importance and relevance of their work. I believe that the work can be published.